# Elimination of Toxic Solvents from Analytical Methods in Food Analysis: Caffeine Determination in Tea as an Example

**DOI:** 10.3390/foods13081189

**Published:** 2024-04-13

**Authors:** Oktawia Kalisz, Aleksandra Jaworska, Sylwia Studzińska, Szymon Bocian

**Affiliations:** Department of Environmental Chemistry and Bioanalytics, Faculty of Chemistry, Nicolaus Copernicus University, 7 Gagarin St., 87-100 Toruń, Poland; oktawia.kalisz@doktorant.umk.pl (O.K.); 302362@stud.umk.pl (A.J.); kowalska@chem.umk.pl (S.S.)

**Keywords:** caffeine, green chemistry, liquid chromatography, solid phase extraction, green chromatography, ethanol, tea

## Abstract

This study presents an innovative method for caffeine determination in tea, employing ethanol as the sole organic solvent for both SPE sample preparation and chromatographic analysis. This approach aligns with green chemistry principles, as confirmed by a comparative study highlighting ethanol’s safety and eco-friendliness compared to traditional solvents. The experiments validate ethanol’s efficacy in caffeine extraction and chromatographic analysis, minimizing environmental impact and eliminating toxicity risks. Utilizing a reduced chromatography column enhances the method’s efficiency and sustainability, resulting in a low limit of quantitation (0.125 μg/mL) and good reproducibility (RSD < 2.5%). Based on tea from the Polish market, the findings reveal the caffeine content (19.29–37.69 mg/g) and endorse ethanol’s role in enhancing sustainable chemical analysis in food science.

## 1. Introduction

Caffeine is a natural psychoactive substance. It belongs to purine alkaloids [1,2,3]. It is found in various plants, such as coffee, tea, cocoa, guarana, and yerba mate, and it is a common ingredient in many beverages [4,5,6]. The most important sources of caffeine are coffee (*Coffea* spp.), tea (*Camellia sinensis*), guarana (*Paullinia cupana*), maté (*Ilex paraguariensis*), cola nuts (*Cola vera*), and cocoa (*Theobroma cacao*) [7,8,9].

An extensively favored and straightforward method for assessing the caffeine levels in various samples involves employing liquid chromatography. This technique facilitates the isolation of caffeine from other compounds present in the sample, enabling its direct identification. Renowned for its precision, this approach is commonly utilized in research facilities. Usually, aqueous extraction (boiling water) is used as the first stage of caffeine determination in tea. Furthermore, various methods are applied to extract caffeine from the water solution. There are known methods that use chloroform for caffeine extraction from water with the addition of lead acetate [7]. The alternative method uses a solid phase extraction (SPE) sample cleanup with a C18 stationary phase [10,11] or other sorbents [12,13,14,15]. Typical solvents include methanol for conditioning and chloroform [10,16] or dichloromethane [11] for elution. Unfortunately, the chloroform and dichloromethane are evaporated to dryness and are consequently released into the atmosphere.

Another problem is that the SPE cartridge must be dried before the sample’s elution with dichloromethane or chloroform. This prolongs the procedure and generates additional costs and organic solvent emissions into the atmosphere to return the sample to a water-miscible solvent for chromatographic analysis. Such solvents should be eliminated from the analytical procedures [17].

SPE methods that do not use chlorinated solvents also exist. In such cases, acetonitrile with 10% acetic acid is used [15] or methanol with 10% acetic acid [13]. This is definitely greener than using chloroform or dichloromethane, but it should be noted that acetonitrile is also a toxic solvent [15].

The most common HPLC methods for caffeine determination in the prepared sample basis on reversed-phase liquid chromatography using C18 stationary phases. As the mobile phase water–acetonitrile or water–methanol mixtures are used, organic solvent content is usually low at around 10–15% but may be as high as 40% or gradient elution is applied [10,18,19,20]. These are the most common organic solvents used in HPLC; unfortunately, both exhibit toxicity [17]. The mobile phase usually contains some additives, such as acetic acid, ammonium acetate, a phosphate buffer [4,11], and even highly toxic trifluoroacetic acid [21].

Currently, the determination of caffeine in tea is not scientifically challenging. Extraction and chromatographic analysis methods were developed many years ago and are widely used [7,22]. One example of a novel methodology may be the automated in-tube solid-phase microextraction coupled with HPLC-ES-MS [23]. The application of ethanol for caffeine extraction from tea was also published; however, ethanol was applied only for extraction. No extract purification on SPE was performed, and the chromatographic procedure was traditional, with acetonitrile and acetic acid as mobile phase components [24].

For this reason, it seems interesting to look at this issue from the point of view of green chemistry and green chromatography. Green liquid chromatography is a form of liquid chromatography designed to reduce its environmental footprint. This is achieved by opting for eco-friendly solvents, materials, and separation techniques, thus promoting sustainability in the process. [21,25,26,27,28].

Standard methods of caffeine determination listed above used chloroform, acetonitrile, or methanol as an organic solvent or organic modifier, which are not green solvents. For this reason, it seems reasonable to develop a method for determining caffeine in tea that is consistent with the principles of green chemistry and green chromatography.

Supercritical fluid extraction methods to extract caffeine from tea leaves are also known. In such cases, ethanol [29,30] or isopropanol [31] is added to carbon dioxide as an organic modifier. This is certainly a step toward green analytical chemistry. Unfortunately, these methods are still not for routine analysis for apparatus reasons.

It is known that ethanol may be applied as an organic modifier for HPLC analysis due to its similar properties to acetonitrile [32,33,34]. Recently, a green method of caffeine determination in dietary supplements was published [35]. For this reason, it is reasonable to ask whether ethanol can also replace organic solvents used in the sample preparation stage, for example, in the SPE technique.

Therefore, this study aimed to develop a method for determining caffeine in tea using ethanol as the only organic solvent used for both sample preparation by the SPE technique and subsequent chromatographic analysis. In addition, the research was carried out on a chromatography column of reduced diameter and length compared to classical HPLC columns. Such a method would be completely compatible with the principles of green chemistry.

## 2. Materials and Methods

### 2.1. Materials and Reagents

The Kromasil Ethernity C18 column (2.1 × 50 mm) with a particle size of 2.5 μm was applied (Nouryon AB, Bohus, Sweden). Water was purified with a Milli-Q Water Purification System (Millipore Corporation, Bedford, MA, USA). HPLC-grade ethanol was obtained from Sigma-Aldrich (Saint Luis, MO, USA). The caffeine standard was purchased from Sigma-Aldrich (Saint Luis, MO, USA). Polymeric SPE cartridges Strata™-X 33 μm (200 mg/3 mL) were purchased from Phenomenex (Torrance, CA, USA).

### 2.2. Instruments

The chromatographic assessments were conducted utilizing the Shimadzu Nexera UHPLC system (Tokyo, Japan). This system features a binary solvent delivery pump (LC-30AD), an autosampler equipped with a 20 μL volume loop (SIL-20AC), a column temperature controller (CTO20AC), and a diode-array UC-detector (SPD-M20A). Data acquisition and instrument control were managed using LabSolution LC/GC 5.65 software developed by Shimadzu in Tokyo, Japan.

### 2.3. Samples

In total, 17 teas available on the Polish market were selected for the study. They were different types of tea: black, green, Pu-erh, Oolong, yellow and white, etc. The kind of tea and country of origin are listed in Table 1.

### 2.4. Methods

The proposed method consists of the hot water extraction of tea material and solid phase extraction for sample purification. Other obtained extracts were analyzed by UHPLC.

#### 2.4.1. Preparation of Sample

A detailed procedure for sample preparation is presented in Figure 1. In detail, a sample of 0.5 g of tea was taken in the beaker, and distilled water (100 mL) was subsequently added. The water temperature was in the range of 90–95 °C according to [36]. After cooling down, the obtained extract was filtered by a nylon filter of 0.45 μm.

Commercial tea consists of many components that cause chromatographic interferences with caffeine. For this reason, the sample preparation consists of SPE with Strata™-X cartridges. It enables the separation of caffeine and removes most of the interfering components.

The SPE adsorbent was conditioned with two portions of 2.5 mL of ethanol and then equilibrated with two portions of 2.5 mL of water. Furthermore, 1 mL of the tea extract was loaded on the cartridge. The washing step was conducted using 2 mL of water. The adsorbed caffeine was eluted with four portions of 0.75 mL ethanol. The obtained extract was diluted with water to 4 mL in a volumetric flask.

The proposed method, in addition to caffeine extraction, also allows the determination of theobromine and theophylline. However, in the case of the studies described here, theobromine and theophylline were not objects of interest.

#### 2.4.2. Chromatographic Method

All measurements were conducted with a flow rate of 0.25 mL/min for the mobile phase. The column temperature was maintained at 40 °C, while the autosampler temperature was set to 5 °C. A volume of 1 μL was injected for each analysis, and detection occurred at 270 nm. Peak purity was checked in the range of 190–800 nm. Unless stated otherwise, measurements were carried out three times to ensure consistency. Separation was achieved using a mobile phase consisting of ethanol and water in a ratio of 10:90 (*v*/*v*). Quantitative analysis was executed utilizing the external standard method.

#### 2.4.3. Method Validation

Various validation parameters of the HPLC method, such as accuracy, precision, linearity, limit of detection (LOD), and limit of quantification (LOQ), were evaluated to confirm the effectiveness of the developed environmentally friendly analytical method.

The LOD and LOQ were established through experimental determination based on the signal-to-noise ratio (LOD = 3*S*/*N* and LOQ = 10*S*/*N*).

Linearity was assessed by generating a calibration curve for caffeine spanning concentrations from 1.25 × 10^−4^ to 0.1 mg/mL. This involved conducting a triple run and plotting the average peak area against the compound concentration to derive the equation.

Precision was evaluated at the following two levels: intraday and inter-day. For intraday precision, three levels of caffeine concentrations (low, medium, and high) were injected in triplicate at three different times within the same day. Inter-day precision involved injecting samples on three separate days in five replicates (1st, 3rd, and 7th day). The reliability of the results was determined by calculating the relative standard deviation (RSD).
$$RSD=\frac{S}{\bar{x}}100\%$$
where $\bar{x}$ is the sample mean and s is the sample standard deviation.

An aqueous caffeine solution with a concentration of 0.25 mg/mL (0.25 mg/g) was prepared and used during the SPE procedure to test the recovery. The SPE procedure was performed in triplicate. After determining the analyte content, the recovery was calculated based on the following Formula (1):(1)$$R=\frac{x}{y}\cdot 100\%$$
where *x*—the determined amount of caffeine in the tested standard sample; *y*—the known amount of caffeine in the tested standard sample.

#### 2.4.4. Greenness of the Method

The greenness of the method was performed according to the AGREEPrep [37] and AGREE calculator [38], into which all required data were entered. Software for them is available for free at [39].

The AGREEPrep calculator evaluates 10 parameters, whereas the AGREE calculator implements 12 factors. Both the AGREE and AGREEprep calculators have assigned weights for each criterion depending on its level of importance from the point of view of green chemistry. The AGREEprep calculator has different weights automatically assigned to all assessed parameters (because the degrees of importance of each criterion vary significantly), and they can be manipulated on a scale from 1 to 5. In the case of AGREE, the starting weight is two, but it can be adjusted to your preferences using a scale from 1 to 4.

## 3. Results and Discussion

### 3.1. Method Development and Validation

A method has been developed to determine caffeine using ethanol as an organic modifier in a reversed-phase liquid chromatography system. The use of ethanol makes the analysis practically and environmentally harmless. However, the question remains whether such a method can be as efficient as traditional methods that use methanol and acetonitrile.

An exemplary chromatogram of tea extract is presented in Figure 2. Firstly, the obtained chromatogram confirms the suitability of the sample preparation using the SPE method. The caffeine peak (3) is fully separated from the other ones. No coeluting impurity was detected in the range of 190–800 nm. Additionally, only peaks of theobromine (1) and theophiline (2) may be found in the chromatogram. In the wavelength range of 190–800 nm and a retention time of more than 0.75 min, no peaks appeared except for caffeine and, eventually, theophylline and theobromine. This indicates that the developed method selectively retains purine alkaloids and allows the removal of other substances found in the tea extract.

Secondly, the mobile phase that consists of 10% EtOH in water effectively separates theobromine, theophylline, and caffeine. It should be stressed that the method was only developed to determine the amount of caffeine in tea. The resulting selectivity and resolution are no worse than traditionally used solvents such as methanol or acetonitrile. Similar results were obtained by Srdjenovic et al.; however, this method uses acetonitrile and tetrahydrofuran [16]. This confirms that using green alternative solvents can offer equally good analytical methods.

It should be noted that the method uses a very short, 5 cm long column. This makes it possible to carry out analyses in a short time (less than 2.5 min). The band broadening is not a problem with the high selectivity and purity of the samples obtained. It also reduces the amount of eluent needed for the analysis, making the analysis greener and more economical than on a classical 4.6 × 150 mm HPLC column.

The following basic validation parameters were determined for the developed UHPLC method: linearity, LOD, LOQ, and inter- and intra-day precision. The method is linear over a wide range of caffeine concentrations from 0.125 μg/mL to 100 μg/mL. Using the regression analysis, the determination coefficient was determined to be equal to 0.9998 (Figure 3). The linearity of the data was also determined using the residual plot. Figure 3 shows that the points are scattered randomly around the zero line, indicating a linear model. Detailed data are listed in Table 2.

It is worth noting that the method’s sensitivity is high (Table 2) compared to the standard HPLC method: 1.9 and 6.3 μg/mL for LOD and LOQ [40] and 0.17 and 0.51 μg/mL for LOD and LOQ [41]. There are also methods with higher sensitivity; however, the separation takes 25 min, and toxic solvents are used [20]. Still, the sensitivity of the developed method can be increased using a smaller particle size column. In the present case, a further increase in sensitivity was not needed because the tea’s caffeine content, and consequently its concentration in the extract, was high. In the case of caffeine determinations in other samples, it is possible to lower the LOQ value. The recovery was 97.87%, which is a test of the high efficiency of the developed sample preparation method.

The precision of the method was determined using intraday and inter-day precision. The results are also listed in Table 2. Only at one point was the RSD higher than 2%. In all other cases, RSD values were lower than 2%, indicating the very high precision of the developed green LC method for caffeine analysis.

### 3.2. Sample Analysis

The caffeine recovery was greater than 97%, and the SPE method showed good recovery and repeatability. It proves that the developed method can be applied to real samples, and for these reasons, it was applied for the extraction of caffeine from tea leaves. Moreover, the developed chromatographic method made it possible to quantify the content of this compound in tea. The results are summarized in Figure 4. The highest caffeine content was observed for yellow tea and the lowest for Pu-erh and some green tea. The caffeine content was relatively similar for the black teas tested, around 25.6 mg/g. A much greater variation was observed for green teas, where the maximum content doubled the lowest content (from 19.29 to 35.27 mg/g). The highest content of caffeine (37.69 mg/g) was found in yellow tea.

The caffeine content was generally within the typical range for tea (2–5%). The obtained values are similar to data reported in the literature., e.g., for black tea from 27 mg/g [7] to 36 mg/g [42]. Generally, the values obtained align with the literature [43,44].

Komes et al. show that caffeine content decreases in the following order: white tea (36.2 mg/g) > yellow tea (31.8 mg/g) > black tea (27.9 mg/g) > oolong tea (27.7 mg/g) > green tea (23.5 mg/g) [7]. Our research shows a different order; however, it may result from the tea source. This is confirmed by the wide variation in green tea’s caffeine content. In the case studied tea in the form of whole leaves showed a significantly higher caffeine content than the more processed forms. In general, studies show that the caffeine content of tea can vary widely [45].

### 3.3. Ecological Aspect of the Developed Method

#### 3.3.1. The Greenness of the Sample Preparation Method

The first criterion, generally of AGREEprep, evaluates the consumption of materials and solvents, as well as energy and time losses. In the developed method, sampling and sample preparation are integrated, which offsets time, energy, and reagent consumption losses. As a result, the method qualifies as in-line/in situ and can be evaluated as the most beneficial from the point of view of green chemistry.

Another criterion evaluates the harmfulness of the solvents and reagents used during the sample preparation process. This also considers the use of acids and bases for derivatization, which should be avoided so as not to generate excessive waste. The developed method uses only distilled water and ethanol. These are non-toxic solvents that do not exhibit bioaccumulation and, therefore, pose no risk to humans or the environment.

The third criterion concerns the use of sustainable or renewable materials during sample preparation. The developed method uses disposable membrane filters and SPE columns in addition to non-sustainable solvents. It has to be underlined that, despite the fact that plastic SPE columns were applied, they can be reused for extraction several times. This reduces waste, although it does not eliminate it. To sum up, only solvents meet the requirements of criterion three, which is less than 50% of all materials used.

Next, the amount of waste generated during the sample preparation step was evaluated. To perform this, the weight of disposable materials (membrane filters), the weight of reusable materials divided by the number of times they were used (SPE columns can be used nine times), and the number of reagents required for the sample preparation (8 mL of ethanol, 7 mL of water and 0.5 g of tea residue) were taken into account. The amount of waste is therefore counted in milliliters rather than microliters, lowering the presented method’s greenness.

The fifth criterion concerns miniaturization and takes into account the size of the samples. Large samples can increase energy requirements, chemical consumption, and waste generation, reducing automation potential. For these reasons, AGREEprep considers samples larger than 100 mL or grams to be unacceptable. In the developed method, only 0.5 g is needed, which is a satisfactorily low amount from the point of view of green chemistry.

Another evaluated parameter is the duration of total sample preparation for analysis. The ideal result in this criterion is the shortest possible preparation of multiple samples in parallel. The developed method uses an SPE chamber that allows the simultaneous extraction of 12 samples, with the extraction of a single sample taking 9 min, making it possible to prepare as many as 80 samples in an hour.

Sample preparation methods typically consist of multiple steps, which can result in the high consumption of materials, energy, and chemicals and wasted time. Striving for the simplicity of operation by integrating steps and automating them are trends in sample preparation that positively impact the environmental performance of the method. It reduces the use of reagents, generates less waste, and reduces human exposure to harmful substances. The developed sample preparation method is as simplified as possible, consisting of preparing tea extracts and carrying out extraction via the SPE technique. Due to the possibility of extracting 12 samples in parallel, it was considered that the procedure could be partially automated.

Criterion eight assesses the desire to make sample preparation energy-saving. To measure this, it is necessary to estimate the total energy consumption per sample and express it in watt-hours. The developed method requires energy consumption to create a vacuum in the SPE chamber. The pump used for this draws 100 W per hour, during which as many as 80 samples can be extracted, so the energy consumption per single sample is very low.

Once properly prepared, samples can be analyzed using various measurements and instrumental techniques. AGREEprep’s ninth criterion suggests choosing simple, energy-efficient techniques and using as few reagents as possible. In the developed method, samples are subjected to chromatographic analysis using ultra-high performance liquid chromatography (UHPLC), which, due to the need to use solvents as the mobile phase, is not considered the most environmentally friendly choice.

The last criterion considers operator safety and environmental hazards, of which only the ethanol used in the procedure should be given special attention due to its flammable properties.

Due to the multiple use of SPE columns and the harmlessness of water and ethanol to the environment, it was decided to reduce the weighting of criteria 3, 4, and 9 by one degree. The results are listed in Figure 5.

#### 3.3.2. The Greenness of the Whole Analytical Procedure

The first parameter of the AGREE calculator relates to the sample preparation stage. The degree of complexity of this stage can be assessed, with direct analytical techniques being the most beneficial and the external treatment of the sample with a large number of steps being the least advantageous. Since samples are prepared manually and placed in a device located nearby and equipped with an automatic sample feeder, the at-line analysis mode was selected. The second criterion evaluated the size and quantity of the samples. The weight of individual samples needed to prepare solutions was only 0.5 g, which is a satisfactory amount according to the AGREE v. 0.5 beta software. The next parameter analyzed the positioning of the analytical device, in which in situ measurements and in-line analysis were the most environmentally friendly. The liquid chromatograph used to analyze the samples was located in the same room as the station where the samples were prepared. Due to the location of the equipment, it was decided to choose at-line analysis. The next criterion assessed the level of integration of analytical processes, indicating that fewer stages were the better. The developed method involved the preparation of extracts, which were then filtered. The analyte was extracted using the SPE technique and then analyzed chromatographically, which is considered green by the AGREE calculator. The sixth parameter takes into account the degree of automation and miniaturization. The high-performance liquid chromatograph used in this procedure was equipped with an autosampler, and its software allowed the analysis of samples with a previously established sequence, thanks to which the procedure met the requirements of being semi-automatic. However, due to apparatus limitations, miniaturization was impossible to achieve. The next criterion evaluated the use of factors responsible for the derivatization process. The developed method does not use this process, so it does not generate additional, potentially harmful waste. The seventh parameter, by evaluating the number of analytes determined during one analysis and its duration, determined the amount of waste generated. The most environmentally friendly solution was to determine many analytes in the shortest possible time. In the developed method, one analyte was determined in 2.5 min, during which only 0.65 mL of the mobile phase was consumed (of which only 0.25 mL was ethanol). According to the AGREE calculator, these are acceptable amounts of waste. Next, in accordance with the principle of reducing energy consumption, the UHPLC device was rated positively. The tenth criterion estimated the number of bio-based reagents. This was met only by water, but the ethanol used in the procedure could also be obtained from renewable sources, which further improved the evaluation of this parameter. The next criterion took into account the use of toxic reagents that did not occur at any stage of the developed method. The use of ethanol as an organic modifier did not negatively affect the evaluation of this parameter since it was used in small amounts and did not pose a threat to the environment. The only negative impact of ethanol occurred in criterion twelve, as ethanol exhibits flammable properties and can pose a danger to the operator.

As with AGREEprep, AGREE benefited from the ability to modify the weighting of individual criteria. To emphasize the small number of steps in the procedure, energy efficiency, and the use of only environmentally friendly solvents, the weights of criteria 4 and 9 were increased by one degree.

The AGREEprep calculator’s evaluation of the developed sample preparation method is listed in Figure 5, and the AGREE calculator’s evaluation of the developed procedure (both sample preparation and chromatographic analysis) is listed in Figure 6. The final ratings are in the middle of the pictograms and are as follows: 0.78 for sample preparation and 0.77 for the analytical procedure. The values obtained correspond to the green color and are close to 1.0, which is the greenest option. This indicates that the developed methods are environmentally friendly.

## 4. Conclusions

The developed method applies an SPE sample preparation and chromatographic analysis with a column that fits the guidelines of green chromatography (shortened column length, reduced particle size) and ethanol as the only organic solvent used. It allows for a short analysis time of less than 2.5 min. The elimination of the toxic organic solvent from the mobile phase during chromatographic analysis and from the sample preparation step causes the proposed method to generate only biodegradable waste. Thus, the resulting procedure confirms that caffeine can be both extracted and determined using only green solvents, ethanol, and water.

The developed method is characterized by a low limit of quantitation equal to 0.125 μg/mL and good reproducibility (relative standard deviation lower than 2.5%). The recovery of the procedure is 97.87%. It confirms that the green analytical method exhibits comparable efficiency to commonly used procedures, whereas its negative impact on the environment is limited. According to AGREE and AGREEprep, the methods developed can be considered environmentally friendly. The results of the calculations were 0.77 and 0.78 for AGREE and AGREEprep, respectively. The pictogram is green, just like the method developed.

The authors intend to encourage analysts to conduct future analyses following green chemistry principles whenever possible. We believe that the results of this paper support this possibility.

## Figures and Tables

**Figure 1 foods-13-01189-f001:**
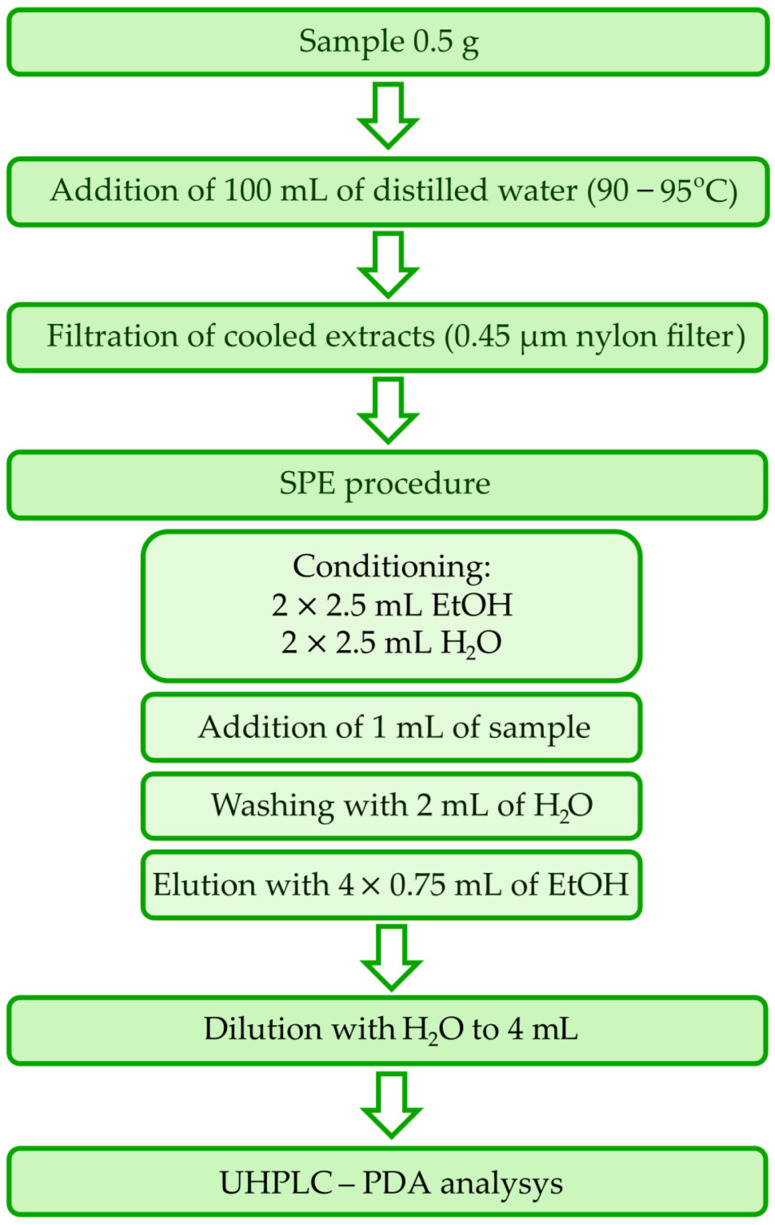
Procedure of sample preparation and analysis.

**Figure 2 foods-13-01189-f002:**
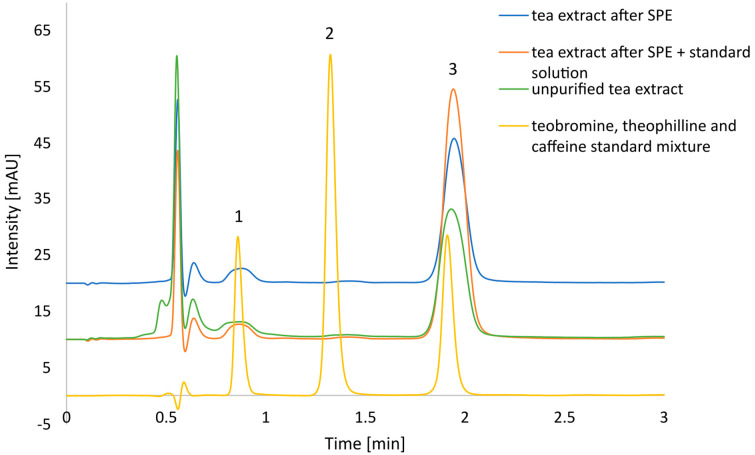
Exemplary chromatogram of theobromine, theophylline, and caffeine mixture and purified tea extract: (1) theobromine, (2) theophylline, and (3) caffeine.

**Figure 3 foods-13-01189-f003:**
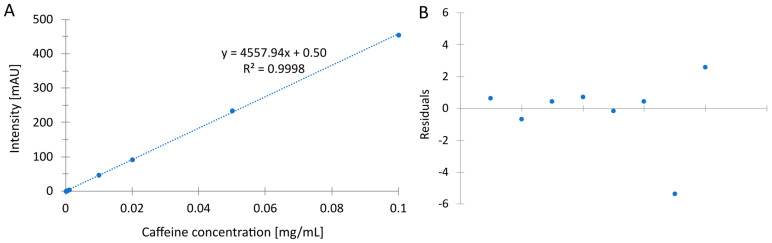
Calibration curve (**A**) and residuals plot (**B**).

**Figure 4 foods-13-01189-f004:**
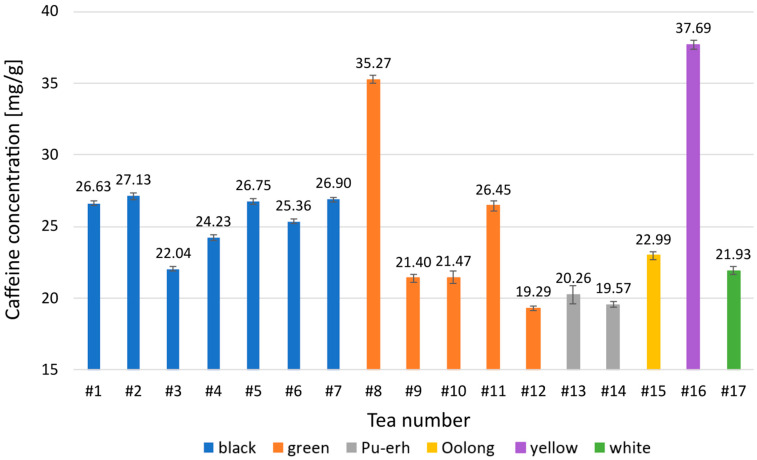
Caffeine contents in analyzed tea samples.

**Figure 5 foods-13-01189-f005:**
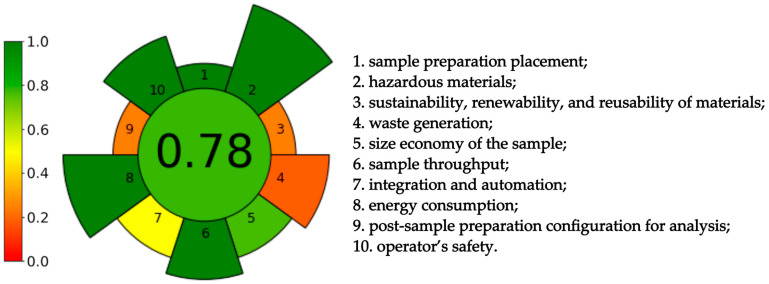
Results of AGREEprep calculation.

**Figure 6 foods-13-01189-f006:**
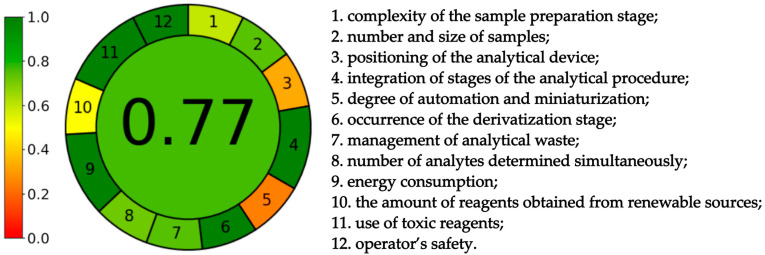
Results of AGREE calculation.

**Table 1 foods-13-01189-t001:** List of tea samples used in the study.

Tea Number	Type of Tea	Country of Origin	Degree of Leaf Fragmentation
#1	black	Sri Lanka	partially ground leaves
#2		Sri Lanka	partially ground leaves
#3		India	granulate
#4		Sri Lanka	granulate
#5		Sri Lanka	granulate
#6		India	granulate
#7		Kenia	granulate
#8	green	China	whole leaves
#9		China	granulate
#10		Sri Lanka	partially ground leaves
#11		Japan	powder
#12		not provided	granulate
#13	Pu-erh	China	partially ground leaves
#14		China	granulate
#15	Oolong	China	whole leaves
#16	yellow	China	whole leaves
#17	white	China	whole leaves

**Table 2 foods-13-01189-t002:** Calibration curve validation parameters and intra- and inter-day precision results.

Calibration Curve Equation	R^2^	LOD [μg/mL]	LOQ [μg/mL]	RSD [%] Intra-Day			RSD [%] Inter-Day			Recovery [%]
				1 [μg/mL]	10 [μg/mL]	50 [μg/mL]	1 [μg/mL]	10 [μg/mL]	50 [μg/mL]	
*y* = 4557.94*x* + 0.50	0.9998	0.06	0.125	1.94	0.43	0.31	2.14	0.73	0.27	97.87 ± 0.65

## Data Availability

The original contributions presented in the study are included in the article, further inquiries can be directed to the corresponding author.

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
