# Peer review of "Elimination of Toxic Solvents from Analytical Methods in Food Analysis: Caffeine Determination in Tea as an Example"

_foods, 2024, doi:10.3390/foods13081189_

Round 1
Reviewer 1 Report
Comments and Suggestions for Authors
The main goal of this research was to study aimed to develop a method for determining caffeine in tea using ethanol as the only organic solvent.
In my point of view, this manuscript is well written with an original or relevant topic within the field.
This article contributes with an innovative method for caffeine determination in tea, employing ethanol as the sole organic solvent and not organic solvent or organic modifier, which are not green solvents (such as chloroform, acetonitrile or methanol).
I would like to state just some information that could be better presented in the text of the manuscript, such as:
Page 4:
The Figure 1 could be better discussed in the text.
Method Validation must present the equations used in the research.
Page 10:
The following sentence in the Conclusions “Allows shorter analysis time.” The shortest analysis time must be presented in this conclusion.
Author Response
Authors thank the Reviewer for taking the time to review our publication. Insightful feedback has greatly enriched the quality of our work.
Page 4:
The Figure 1 could be better discussed in the text.
The description in part 2.4.1 was corrected.
Method Validation must present the equations used in the research.
Equation for RSD was added
Page 10:
The following sentence in the Conclusions “Allows shorter analysis time.” The shortest analysis time must be presented in this conclusion.
It is corrected
Reviewer 2 Report
Comments and Suggestions for Authors
The introduction lacks reference to methods similar to the one developed (using SPE for purification of tea extract and HPLC UV).
Line 31: “There are known methods (…)” – there should be any reference after this sentence.
Section 2.4.2.: “The detection was performed at 270 nm.” When using the DAD detector, was peak purity checked in any way? This is also related to the statement in line 169: "Caffeine peak (3) is fully separated from other ones." Drawing such a conclusion solely based on the appearance of the chromatogram (obtained for one wavelength!) does not provide any certainty regarding the lack of coelution with other sample components. Similarly, the following sentences in lines 169-172 contain statements that are not supported by the results (although the use of the DAD detector would provide such possibilities to some extent). A similar situation concerns the statement in line 183: "(...) high selectivity and purity of the samples obtained."
Chapter 2.4.3. Method validation, to line 128: the description shows that validation concerns the evaluation of the HPLC method itself, and not - as the authors say: "analytical procedure". This should at least be mentioned in the publication, although it would be most appropriate to characterize the entire method, not just the final analysis (HPLC). N.B. line 127 “(…) the samples were injected (…)” – no information about the sample type.
Line 129 and next: the procedure for determining recovery is based on the use of an aqueous solution for experiments. Unfortunately, no real samples have been tested, the components of which may have an impact on the SPE stage. The use of solutions prepared in pure water seems to be too far-reaching a simplification.
Line 129: “(…) concentration of 0.25 mg/mL (…)": please present the concentration also in mg/g units
Section 2.4.4: There is little point in listing the factors considered in the AGREEPrep and AGREE calculators (lines 140-153) when they are described in section 3.3. The best place to provide them is the legend in Figure 5.
Line 197: “(…) sensitivity is high (Table 2)” – what was the basis for such a conclusion? Only a comparison with literature data on the slope coefficients of calibration lines for a similar concentration range could make it possible to draw such a conclusion. Nb: in Table 2, “x” is missing from the equation.
Chapter 3.2. – no comparison of the results obtained with the works of other authors.
Chapters 3.3.1. and 3.3.2. contain unnecessary general descriptions of the individual factors taken into account in the calculators used (for example, lines 155-260). These chapters should be significantly shortened (maybe in tabular form?). There are also no literature references and no comparison of the results obtained with the work of other researchers
Line 175-176: statement not supported by a literature reference.
Lines 349-351: in their current form, these are not conclusions: instead of this postulate, the conclusions from chapter 3.3 should be included.
Referring to the iThenticate report indicates problems mainly related to the description contained in the experimental part), I propose rewording fragments of the text taken from previous publications.
Comments on the Quality of English LanguageCorrect the style (the form of the statement) of some sentences. SOME examples:
Line 39-43: This sentence is too complex and too long
Line 83: “They are different teas, black, green etc.”
Line 89-90: “Further obtained extracts are analyzed by UHPLC”
Line 96-97: “(…) the sample preparation proposed consists of SPE with Strata-X cartridges.”
Figure 1. Inconsistent grammatical forms of verbs are used in the table i.e.: infinitives such as "filtration" and imperatives such as "add" - this should be unified.
Line 109: “(…) the mobile phase’s 0.25 mL/min flow rate.” – the Saxon genitive (phase’s) is used mainly to refer to people.
Lines 181-184: This paragraph uses comparative adjectives (shorter, greener) and other similarly significant phrases - but there is no reference to what the forms of description used should be referred to. Similar situation in line 65.
Author Response
Thank you very much for this review. The critical comments have significantly improved the quality of this work.
The introduction lacks reference to methods similar to the one developed (using SPE for purification of tea extract and HPLC UV).
- Additional literaturÄ™ was provided.
Line 31: “There are known methods (…)” – there should be any reference after this sentence.
- Literature is provided.
Section 2.4.2.: “The detection was performed at 270 nm.” When using the DAD detector, was peak purity checked in any way? This is also related to the statement in line 169: "Caffeine peak (3) is fully separated from other ones." Drawing such a conclusion solely based on the appearance of the chromatogram (obtained for one wavelength!) does not provide any certainty regarding the lack of coelution with other sample components. Similarly, the following sentences in lines 169-172 contain statements that are not supported by the results (although the use of the DAD detector would provide such possibilities to some extent). A similar situation concerns the statement in line 183: "(...) high selectivity and purity of the samples obtained."
- We agree with the reviewer's concerns. Peak purity was checked in the range 190-800 nm. No coeliuting impurity was detected.
Chapter 2.4.3. Method validation, to line 128: the description shows that validation concerns the evaluation of the HPLC method itself, and not - as the authors say: "analytical procedure". This should at least be mentioned in the publication, although it would be most appropriate to characterize the entire method, not just the final analysis (HPLC). N.B. line 127 “(…) the samples were injected (…)” – no information about the sample type.
- We agree with the reviewer's concerns. The description of validation was corrected.
Line 129 and next: the procedure for determining recovery is based on the use of an aqueous solution for experiments. Unfortunately, no real samples have been tested, the components of which may have an impact on the SPE stage. The use of solutions prepared in pure water seems to be too far-reaching a simplification.
- We agree with the reviewer's concerns. The best solution, of course, would be to use certified reference material, however, we do not have such. For this reason, we have adopted this strategy. We hope that it is acceptable.
Line 129: “(…) concentration of 0.25 mg/mL (…)": please present the concentration also in mg/g units
- It is aqyeous solution, so the concentration is 0.25 mg/g. It is added to the manuscript.
Section 2.4.4: There is little point in listing the factors considered in the AGREEPrep and AGREE calculators (lines 140-153) when they are described in section 3.3. The best place to provide them is the legend in Figure 5.
- Figure 5 was modified. Additional figure 6 is provided. Description is added to figures.
Line 197: “(…) sensitivity is high (Table 2)” – what was the basis for such a conclusion? Only a comparison with literature data on the slope coefficients of calibration lines for a similar concentration range could make it possible to draw such a conclusion.
- Literature data are proivided for comparison
Nb: in Table 2, “x” is missing from the equation.
- Corrected
Chapter 3.2. – no comparison of the results obtained with the works of other authors.
- Thank you very much for this comment. Thanks to it, we were able to find an error in our calculations. The calculations have been righted and the results compared with the literature.
Chapters 3.3.1. and 3.3.2. contain unnecessary general descriptions of the individual factors taken into account in the calculators used (for example, lines 155-260). These chapters should be significantly shortened (maybe in tabular form?). There are also no literature references and no comparison of the results obtained with the work of other researchers
- The AGREE calculator is relatively new, and if possible we would like to keep this description. We think this is a good instruction for other researchers on how to operate AGREE and AGREEprep
Line 175-176: statement not supported by a literature reference.
- Literature is provided.
Lines 349-351: in their current form, these are not conclusions: instead of this postulate, the conclusions from chapter 3.3 should be included.
- Conclusion were corrected according to part 3.3. We would like to keep the postulate at the end of the conclusions, if the Reviewer agree.
Referring to the iThenticate report indicates problems mainly related to the description contained in the experimental part), I propose rewording fragments of the text taken from previous publications.
- Similar paragraphs were rephrased.
Comments on the Quality of English Language
Correct the style (the form of the statement) of some sentences. SOME examples:
Line 39-43: This sentence is too complex and too long
- Corrected
Line 83: “They are different teas, black, green etc.”
- Corrected
Line 89-90: “Further obtained extracts are analyzed by UHPLC”
- Corrected
Line 96-97: “(…) the sample preparation proposed consists of SPE with Strata-X cartridges.”
- Corrected
Figure 1. Inconsistent grammatical forms of verbs are used in the table i.e.: infinitives such as "filtration" and imperatives such as "add" - this should be unified.
- Figure is corrected.
Line 109: “(…) the mobile phase’s 0.25 mL/min flow rate.” – the Saxon genitive (phase’s) is used mainly to refer to people.
- Corrected
Lines 181-184: This paragraph uses comparative adjectives (shorter, greener) and other similarly significant phrases - but there is no reference to what the forms of description used should be referred to. Similar situation in line 65.
- Corrected
Reviewer 3 Report
Comments and Suggestions for Authors
The manuscript is well written and contributes to green chemistry in this circular economy. The use of green solvents, such as ethanol, in analysis is commendable.
The experimental methodology is well described to allow replication of the study and is adjudged to be adequate. However, the section on sampling would have been improved by including the sampling strategy adopted. Also, there is need to give justification for sample preparation, especially the use of water in the temperature range of 90–95 °C
Comments on the Quality of English Language
The manuscript is well written and contributes to green chemistry in this circular economy. The use of green solvents, such as ethanol, in analysis is commendable.
The experimental methodology is well described to allow replication of the study and is adjudged to be adequate. However, the section on sampling would have been improved by including the sampling strategy adopted. Also, there is need to give justification for sample preparation, especially the use of water in the temperature range of 90–95 °C
Author Response
We thank the reviewer for his comments, which improved the quality of this work.
The experimental methodology is well described to allow replication of the study and is adjudged to be adequate. However, the section on sampling would have been improved by including the sampling strategy adopted.
- The primary goal of the work was to develop an ecological method for determining caffeine. The selection of samples was not specifically designed. We wanted to test our green method on real samples.
Also, there is need to give justification for sample preparation, especially the use of water in the temperature range of 90–95 °C.
- A literature reference for the recommended procedure is provided.
Reviewer 4 Report
Comments and Suggestions for Authors
The manuscript describes the adaptation of analytical methods, using the determination of caffeine in tea as an example, to comply with the green chemistry principles.
The aim of the study and the results are well-described. The argumentations in the introduction section are particularly interesting.
Overall, this work is a valuable contribution to research on the improvement of analytical methods and the use of less harmful solvents.
Please find below my specific comments.
Line 123: I am aware that caffeine is a drug, but particularly in this manuscript dealing with its detection in tea, I would recommend replacing “drug” with another word such as “compound” or similar. When “drug “ is used, it may appear to readers as if teas are direct sources of drugs, while caffeine is just an inherent component of tea.
Lines 127-128: It is not clear what “1st, 3rd, and 7th” refers to. Please clarify.
Lines 167-172: According to the manuscript, Figure 2 displays the suitability of the sample preparation by SPE. It is not clear how this figure demonstrates that, as it presents a mixture of the 3 target compounds (possibly dissolved in a neat solvent, as it is not specified for the chromatogram in yellow) and purified tea extract (chromatogram in blue). Is the peak around 1.8 min on the “blue” chromatogram caffeine, or is it an interference? To better demonstrate the efficiency of the SPE, a comparison is made between unpurified extract and extract purified by SPE, spiked and unspiked with the target analytes. Please clarify or adapt the figure.
Author Response
We thank the reviewer for his comments, which improved the quality of this work.
Line 123: I am aware that caffeine is a drug, but particularly in this manuscript dealing with its detection in tea, I would recommend replacing “drug” with another word such as “compound” or similar. When “drug “ is used, it may appear to readers as if teas are direct sources of drugs, while caffeine is just an inherent component of tea.
- We agreed with the Reviewer. It is corrected.
Lines 127-128: It is not clear what “1st, 3rd, and 7th” refers to. Please clarify.
- It is corrected.
Lines 167-172: According to the manuscript, Figure 2 displays the suitability of the sample preparation by SPE. It is not clear how this figure demonstrates that, as it presents a mixture of the 3 target compounds (possibly dissolved in a neat solvent, as it is not specified for the chromatogram in yellow) and purified tea extract (chromatogram in blue). Is the peak around 1.8 min on the “blue” chromatogram caffeine, or is it an interference? To better demonstrate the efficiency of the SPE, a comparison is made between unpurified extract and extract purified by SPE, spiked and unspiked with the target analytes. Please clarify or adapt the figure.
- We agreed with the Reviewer. A new figure was prepared that contains more data.